# Interseismic Fault Coupling and Slip Rate Deficit on the Central and Southern Segments of the Tanlu Fault Zone Based on Anhui CORS Measurements

Tingye Tao [†], Hao Chen [†], Shuiping Li *, Xiaochuan Qu and Yongchao Zhu

College of Civil Engineering, Hefei University of Technology, Hefei 230009, China; taotingye@hfut.edu.cn (T.T.); haochen@mail.hfut.edu.cn (H.C.); qqxxcc@hfut.edu.cn (X.Q.); yczhu@hfut.edu.cn (Y.Z.)
* Correspondence: lishuiping@hfut.edu.cn; Tel.: +86-1582-717-2250
† These authors contributed equally to this work.

**Abstract:** The Tanlu fault zone, extending over 2400 km from South China to Russia, is one of the most conspicuous tectonic elements in eastern Asia. In this study, we processed the Global Positioning System (GPS) measurements of Anhui Continuously Operating Reference System (AHCORS) between January 2013 and June 2018 to derive a high-precision velocity field in the central and southern segments of the Tanlu fault zone. We integrated the AHCORS data with those publicly available for geodetic imaging of the interseismic coupling and slip rate deficit distribution in the central and southern segments of the Tanlu fault zone. This work aims at a better understanding of strain accumulation and future seismic hazard in the Tanlu fault zone. The result indicates lateral variation of coupling distribution along the strike of the Tanlu fault zone. The northern segment of the Tanlu fault zone has a larger slip rate deficit and a deeper locking depth than the southern segment. Then, we analyzed three velocity profiles across the fault. The result suggests that the central and southern segments of the Tanlu fault zone are characterized by right-lateral strike-slip (0.29–0.44 mm/y) with compression components (0.35–0.76 mm/y). Finally, we estimated strain rates using the least-squares collocation method. The result shows that the dilatation rates concentrate in the region where the principal strain rates are very large. The interface of extension and compression is always accompanied by sudden change of direction of principal strain rates. Especially, in the north of Anhui, the dilatation rate is largest, reaching $3.780 \times 10^{-8}/a$. Our study suggests that the seismic risk in the northern segment of the Tanlu fault zone remains very high for its strong strain accumulation and the lack of historical large earthquakes.

**Keywords:** Anhui CORS; velocity field; fault coupling; slip rate deficit; strain rates

## 1. Introduction

Tanlu fault zone, which is mainly characterized by right-lateral strike-slip and reverse components, is one of the largest fault zones in eastern China [1,2]. It runs from Heilongjiang Province in the north to the shore of the Yangtze River, Hubei Province, in the south, totaling 2400 km. The Tanlu fault zone crosses multiple tectonic blocks from north to south and its internal structure is complex. Many earthquakes with magnitudes larger than $M_s$ 5.0 have occurred in the Tanlu fault zone since the Quaternary [3]. For instance, in 1668, an $M_s$ 8.5 earthquake occurred in Tancheng, located on the Yishu fault of the central segment of the Tanlu fault zone, resulting in huge casualties and property losses [4,5]. Additionally, the Tanlu fault zone has also been influenced by the far-field post-seismic effects of many large earthquakes during the past decades, such as the 2008 Wenchuan $M_s$ 8.0 earthquake and the 2011 $M_w$ 9.0 Tohoku, Japan Earthquake [6–8]. The future seismic risk for this densely populated region is still not very clear due to the lack of a detailed study of the crustal deformation and strain accumulation in the Tanlu fault zone. The seismogenic potential on the Tanlu fault zone is largely dominated by the mechanical

properties on the fault interface, in which the coupling ratio (also inferred as locking degree) and slip rate deficit are two vital indicators that can shed light on the strain buildup on fault interfaces. Many previous studies have explored the interseismic coupling and estimated the slip rate deficit on a series of active faults using the Global Positioning System (GPS) and Interferometric Synthetic Aperture Radar (InSAR) measurements [9–11]. For instance, Zhao et al. (2017) inverted the fault locking and fault slip deficit in the main Himalaya thrust fault [12]. They have shown that the maximal magnitude and the rupture extents of large earthquakes on a fault can be well constrained by the spatial distribution of interseismic coupling. Therefore, a refined coupling image for the Tanlu fault zone could help us to assess the strain buildup and future earthquake hazard in this region.

The characteristics of crustal movement and strain accumulation of the Tanlu fault zone have been constrained by geodetic surveying; most of them are GPS observations [13–15]. Recently, Li et al. (2020) used the two periods of GPS horizontal velocity field in the north China between 1999–2017 to invert the fault locking and slip rate deficit of the Tanlu fault zone, by utilizing the back-slip dislocation model, and compared the differences between two periods [16]. They found that the 2011 Tohoku $M_W$ 9.0 earthquake played a vital role in alleviating the strain accumulation of the Tanlu fault zone. Li et al. (2016) inverted the fault locking and slip rate deficit of the Tanlu fault zone by using GPS horizontal velocity of 2009–2014 in North China and verified that different reference frames have little impact on the inversion results [7]. They suggested that the locking depth in the northern end of the Tanlu fault zone is nearly 27 km depth, while the locking depth reaches 32 km in the central segment and then it decreases to only 5 km in the southern end of the Tanlu fault zone, exhibiting lateral variation of fault locking along the strike of the Tanlu fault zone.

It can be inferred that the first-order characteristics of the fault activity and strain buildup on the Tanlu fault zone have been constrained by previous geodetic surveying. However, we still found that fault slip and interseismic coupling distribution on each segment of the Tanlu fault zone revealed by different studies show significant discrepancies. We attributed them to various timescales of GPS data and different parameter settings in the modeling. Additionally, the spatial resolution of interseismic coupling image on the Tanlu fault zone remains low due to the lack of near-field GPS observation. With the completion of the Anhui Continuously Operating Reference System (AHCORS) in 2011 and upgrading in 2016, there are more than 50 AHCORS stations throughout the Anhui Province. Most of these stations are located in the near-field of the central and southern segments of the Tanlu fault zone, providing a valuable chance for us to further study the motion characteristics and strain accumulation in the Tanlu fault zone. In this paper, we processed GPS data of 50 AHCORS stations from January 2013 to June 2018 with time period of nearly 6 years. In combination with the data of Crustal Movement Observation Network of China (CMONOC) between 1999–2016 computed by Wang (2020) [17], we obtained a complete velocity field of the central and southern segments of the Tanlu fault zone. This new velocity field is then employed to invert for the fault locking and slip rate deficit on the central and southern segments of the Tanlu fault zone. Finally, we discussed the characteristics of the fault slip according to two-dimensional velocity analysis. We also analyzed the strain accumulation based on the least-squares collocation method, which has the advantage that the higher the density of the site distribution, the higher the accuracy. Our work provides useful constraints on the fault slip motion and sheds new light on the seismic hazard of the central and southern segments of the Tanlu fault zone.

The rest of the paper is organized as follows. In Section 2, the tectonic setting of the study region is described. In Section 3, datasets, methodology, and fault geometry are presented. Section 4 demonstrates the results of fault coupling ratios, fault slip rate deficit, and velocity profiles. Section 5 discusses how to use checkboard tests to assess the spatial resolution of the fault coupling ratios, comparison with previous studies, strain characteristics, and implication for seismic hazard.

## 2. Tectonic Setting

The Tanlu fault zone represents a NE-trending continental-scale strike-slip fault zone with high levels of seismicity in East China and records the evolutionary history of plate interactions in East Asia during Mesozoic and Cenozoic times [18]. The Tanlu fault zone is commonly interpreted to have generated due to the collision between the North China block and Yangtze Plate from the middle Triassic [19]. After that, the fault zone was transformed to an extensional structure by the Cretaceous–Paleocene that controlled several grabens filled by Cretaceous volcanic rocks and terrestrial clasts [20]. In terms of geographical location, the Tanlu fault zone is always divided into three segments, that are the northern segment in northeast China, the central segment in the Bohai Bay, and the southern segment from Shandong Province to Anhui Province. The central and southern segments of the Tanlu fault zone, which act as the boundary faults that separate the North China block from the Subei basin and the Sulu belt, experienced a complex deformation characterized by Mesozoic sinistral and Cenozoic dextral motions. The rock types across the fault zone change abruptly from Archean to Paleoproterozoic high-grade metamorphic basement rocks in the North China block to ultra-high-pressure metamorphic rocks and Mesozoic granites in the Sulu belt [21]. The fault outcrops along the central and southern segments of the Tanlu fault zone are diffusely distributed. The central segment of the Tanlu fault zone is composed of five subparallel faults [20]. The fault activities of these faults remain controversial. Both reverse and dextral slips have been proposed to explain the Quaternary activities of these subparallel faults [20,22]. Many large earthquakes have occurred in this segment, such as the 1668 Tancheng Ms 8.5 earthquake and the 1969 Mw 7.4 Bohai earthquake, suggesting that the central segment is still an active earthquake zone [23]. In contrast, the fault activity of the southern segment of the Tanlu fault zone is much weaker than the central segment, consistent with the weak seismic activity of the southern segment [22].

## 3. Data and Methods

### 3.1. Data Processing

The GPS data we collected are mainly from AHCORS between January 2013 and June 2018. Figure 1 shows the spatial distribution of AHCORS stations. All the GPS data are processed using the GAMIT/GLOBK software (Ver.10.7) with a double-difference approach to generate daily solutions [24]. In the processing, we eliminated the GPS stations with data integrity less than 90 percent. More than nine International GNSS Service (IGS) stations around the Chinese mainland are adopted in the processing.

In the detailed processing strategy, we adopt the Vienna Mapping Function 1 (VMF1) to correct the tropospheric delay, and a zenith wet delay parameter is estimated every 2 h [25]. The most recent global ocean tide model (Finite Element Solutions 2004, FES2004) is used to correct the station displacements induced by ocean tides [26]. The detailed geophysical models and parameter settings used in the processing are listed in Table 1.

The specific processing is as follows. Firstly, GAMIT is used to obtain daily solutions that are loosely constrained for station coordinates and satellite orbits. Secondly, the global H files released by Scripps Orbits and Permanent Array Center (SOPAC) are used for network adjustment. Finally, the velocity field of AHCORS stations in the International Terrestrial Reference Frame 2008 (ITRF2008) can be obtained through coordinate frame transformation. For the convenience of tectonic interpretation, we transformed the horizontal velocity field from ITRF2008 to a stable Eurasia frame using Euler vectors for the Eurasian plate proposed by Wang et al. (2020) (−0.087, −0.514, and 0.741 mas/a) [17], and the velocity field is listed in Table 2. Figure 2 displays the horizontal velocity field on the central and southern segments of the Tanlu fault zone and its surrounding areas under the Eurasia reference frame, including the velocity field of AHCORS calculated by ourselves and the velocity field of CMONOC. Generally, the velocity of AHCORS stations coincides well with the velocity of CMONOC stations, and our AHCORS stations show higher precision thanks to the longer observation time.

**Table 1.** Data processing strategy.

| Data Processing Strategy | Option |
|---|---|
| Sampling interval | set sint = '30' |
| Number of epochs | set nepc = '2880' |
| Start time for processing | set stime = '0 0' |
| Choice of Experiment | RELAX. |
| Type of Analysis | 1-ITER |
| Choice of Observable | LC_AUTCLN |
| Zenith Delay Estimation | Y |
| Met obs source | GPT 50 |
| DMap | VMF1 |
| WMap | VMF1 |
| Use otl.grid | Y |
| Use atml.grid | Y |
| Use atl.grid | Y |

**Table 2.** Site velocity solution.

| Station | Ve_I [a] | Ve_E [b] | dVe [c] | Vn_I [d] | Vn_E [e] | dVn [f] |
|---|---|---|---|---|---|---|
| AQSS | 33.831 | 7.488 | 0.234 | −10.813 | −1.414 | 0.168 |
| AQYX | 33.672 | 7.353 | 0.166 | −11.961 | −2.509 | 0.120 |
| BZGY | 34.786 | 8.504 | 0.161 | −9.981 | −0.568 | 0.155 |
| BZLX | 34.122 | 7.830 | 0.148 | −10.666 | −1.254 | 0.127 |
| BZMC | 33.767 | 7.509 | 0.157 | −11.812 | −2.325 | 0.143 |
| CHCH | 33.850 | 7.669 | 0.155 | −12.460 | −2.677 | 0.104 |
| CHJU | 33.475 | 7.249 | 0.152 | −12.544 | −2.890 | 0.105 |
| CZDY | 33.518 | 7.346 | 0.136 | −11.340 | −1.590 | 0.110 |
| CZLA | 33.876 | 7.769 | 0.150 | −12.224 | −2.311 | 0.120 |
| CZMG | 33.769 | 7.632 | 0.153 | −12.219 | −2.400 | 0.130 |
| CZQJ | 33.485 | 7.358 | 0.149 | −11.351 | −1.465 | 0.128 |
| CZQY | 36.731 | 10.539 | 0.204 | −11.237 | −1.446 | 0.129 |
| CZST | 34.930 | 8.702 | 0.175 | −12.303 | −2.595 | 0.128 |
| CZTC | 33.338 | 7.294 | 0.149 | −11.851 | −1.808 | 0.128 |
| CZZT | 35.710 | 9.424 | 0.205 | −12.047 | −2.490 | 0.097 |
| FYFN | 33.736 | 7.380 | 0.147 | −11.544 | −2.265 | 0.120 |
| FYFY | 34.555 | 8.217 | 0.141 | −10.189 | −0.879 | 0.121 |
| FYJS | 35.317 | 8.953 | 0.147 | −10.143 | −0.924 | 0.119 |
| FYLQ | 34.635 | 8.261 | 0.155 | −9.803 | −0.595 | 0.125 |
| FYTH | 32.688 | 6.348 | 0.141 | −11.228 | −1.939 | 0.113 |
| FYYS | 32.883 | 6.583 | 0.158 | −9.438 | −0.014 | 0.115 |
| HFCF | 33.092 | 6.871 | 0.161 | −11.345 | −1.712 | 0.124 |
| HFFD | 34.026 | 7.820 | 0.170 | −11.239 | −1.535 | 0.138 |
| HSHS | 33.885 | 7.713 | 0.176 | −11.645 | −1.793 | 0.130 |
| HSQM | 32.905 | 6.693 | 0.174 | −11.953 | −2.196 | 0.123 |
| LAHS | 33.551 | 7.237 | 0.164 | −11.113 | −1.666 | 0.130 |
| LALA | 33.863 | 7.569 | 0.145 | −10.967 | −1.480 | 0.099 |
| MASM | 33.091 | 6.979 | 0.156 | −11.844 | −1.899 | 0.133 |
| SZDS | 34.521 | 8.285 | 0.140 | −14.830 | −5.385 | 0.122 |
| SZSX | 32.867 | 6.741 | 0.156 | −10.695 | −0.902 | 0.143 |
| SZXX | 33.922 | 7.736 | 0.159 | −12.062 | −2.473 | 0.125 |
| XCGD | 33.422 | 7.372 | 0.167 | −12.799 | −2.664 | 0.127 |
| XCJD | 33.703 | 7.566 | 0.167 | −9.387 | 0.553 | 0.125 |
| XCJN | 33.578 | 7.434 | 0.172 | −12.919 | −3.008 | 0.125 |
| XCJX | 33.686 | 7.551 | 0.178 | −11.446 | −1.494 | 0.128 |
| XCLX | 34.776 | 8.710 | 0.164 | −12.279 | −2.195 | 0.121 |
| XCNG | 34.176 | 8.082 | 0.181 | −12.847 | −2.809 | 0.130 |

[a] East components under ITRF2008 reference frame. [b] East components under Eurasian plate. [c] East velocity uncertainties. [d] North components under ITRF2008 reference frame. [e] North components under Eurasian plate. [f] North velocity uncertainties.

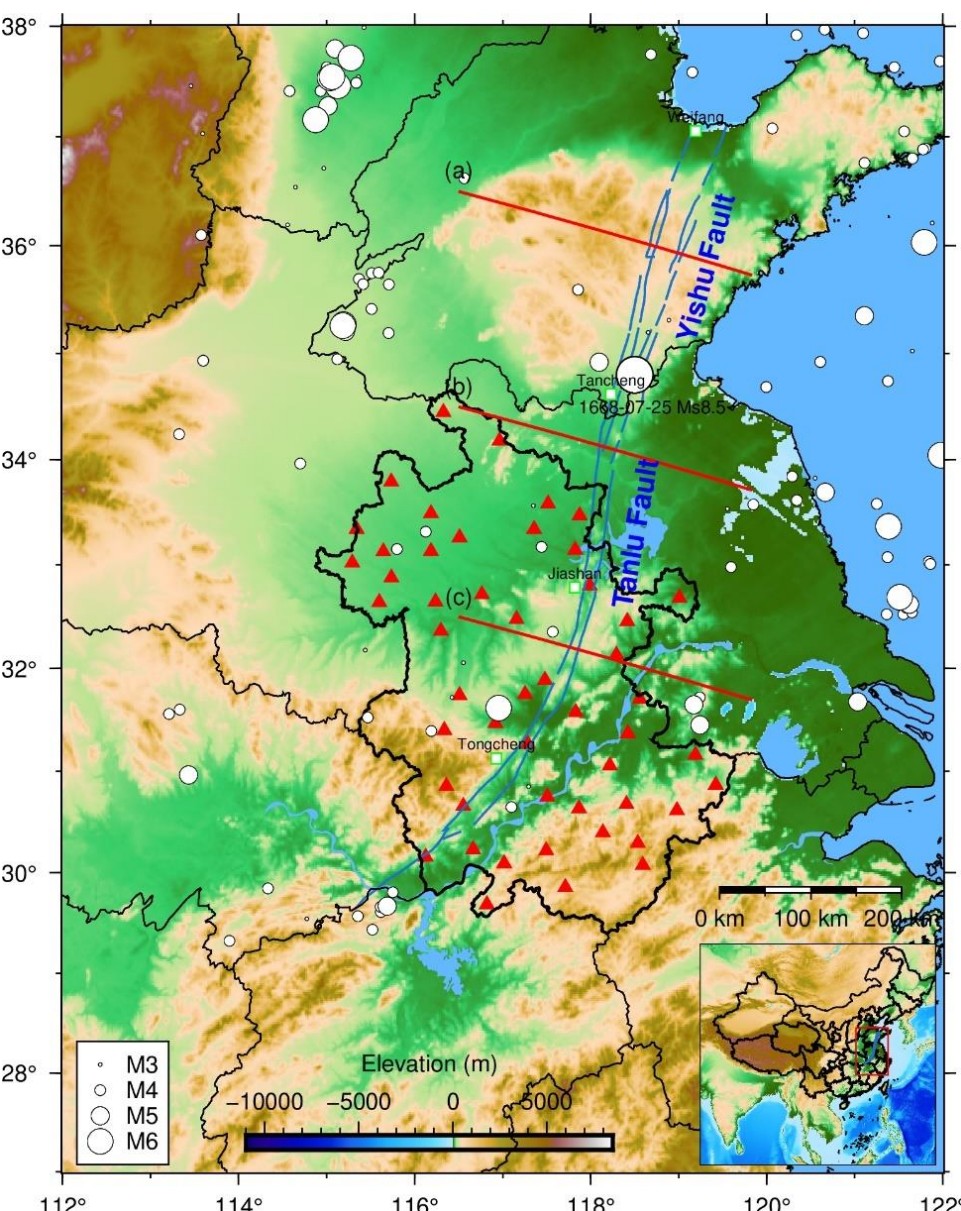

**Figure 1.** Distribution map of Anhui CORS stations. The red triangles are where the Anhui CORS stations are located. White circles show earthquakes with $3.0 \leq M_s \leq 8.5$ from 1 January 1900 to 31 December 2020 (https://earthquake.usgs.gov, accessed on 14 December 2021). The largest white circle is an $M_s$ 8.5 earthquake that struck Tancheng on 25 July 1668. The blue broken line represents the central and southern segments of the Tanlu fault zone, and the Yishu fault zone is a part of the Tanlu fault zone in Shandong Province. The three red lines labeled by (**a**–**c**) represent the locations of three velocity profiles across the Tanlu fault zone.

### 3.2. Modeling Approach

In this study, we adopted the Fortran-based DEFNODE software [27,28] for the inversion. The DEFNODE software has been widely used to invert the interseismic block rotation, fault locking, and slip deficit in the northwestern U.S., Pacific Northwest, and southern Cascadia [29–31].

The DEFNODE program assumes that the movement of the points in the blocks are the sum of the surface elastic deformation caused by the block rotation, the uniform strain rates within blocks, and slip deficit at the block boundary due to fault locking. Constrained by GPS vector, surface uplifts, earthquake slip vector, spreading rates, or other data, we

can use grid searches or simulated annealing technique to estimate fault coupling ratios, fault slip rates, and Euler pole at the block boundary. McCaffrey (2002) [32] proposed the expression if there are no uniform strain rates within blocks:

$$V_i(X) = \sum_{b=1}^{B} H(X \in \Delta_b)[_R\Omega_b] \cdot i - \sum_{k=1}^{F} \sum_{n=1}^{N_k} \sum_{j=1}^{2} \Phi_{nk} G_{ij}(X, X_{nk}) \left[ _h\Omega_f \times X_{nk} \right] \cdot j \qquad (1)$$

Where $X$ is the position of GPS stations, $B$ is the number of blocks, $\Delta_b$ is the area range of block $B$ ($H = 1$ if the station is within the range of block $B$, otherwise $H = 0$), $i$ is the unit vector in the $i$ direction, and $_R\Omega_b$ is the Euler rotation pole of block $B$ with respect to the reference frame. $_h\Omega_f = {}_R\Omega_b - {}_f\Omega_R$ is the Euler vector of footwall block $f$ relative to the hanging wall block, F is the number of faults, $N_k$ is the number of nodes for fault $k$, $j$ is the unit vector of direction $j$ on the fault surface, $\Phi_{nk}$ is the fault coupling ratio of node $n$ on fault $k$, and $X_{nk}$ is the position of node $n$ on fault k. For $G_{ij}(X, X_{nk})$, it represents the response function of the velocity of surface point $X$ in the direction of $i$ generated by unit slip in the $j$ direction at node $X_{nk}$ on the fault. $V$ is velocity, and the unit of V is in millimeters per year.

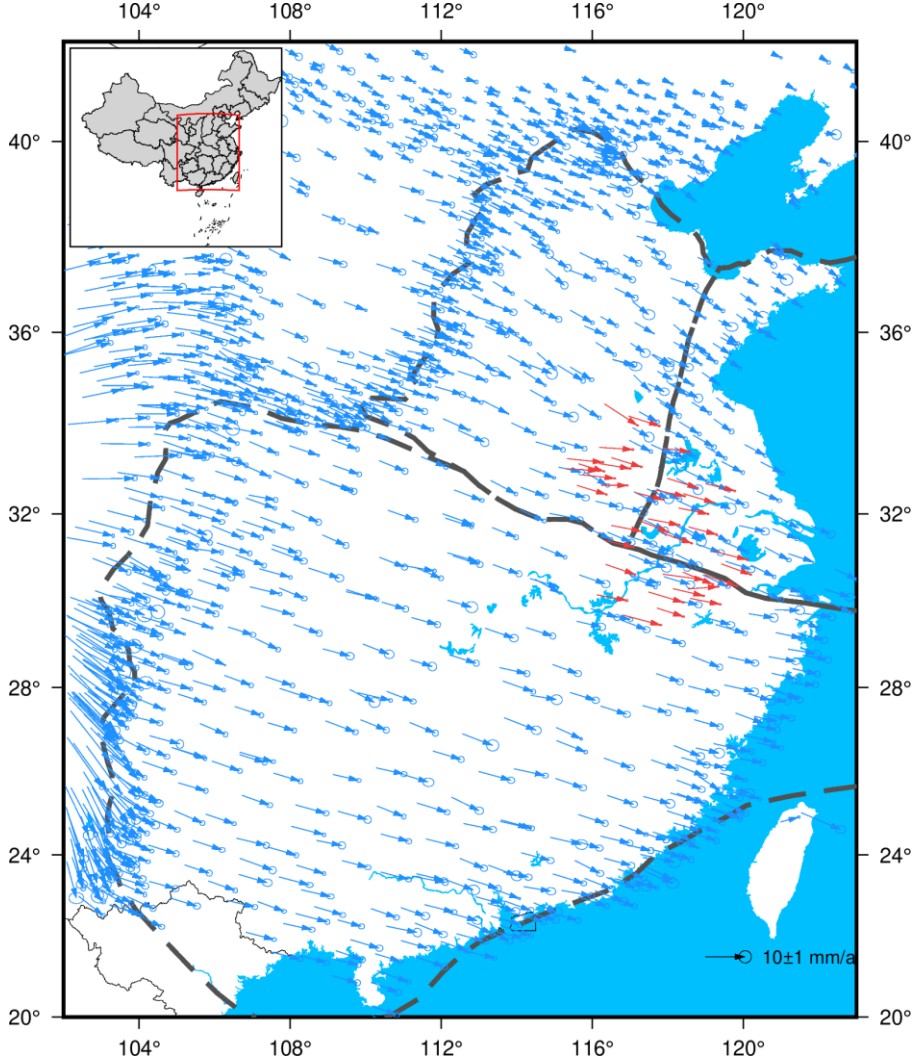

**Figure 2.** GPS velocity field with respect to Eurasian plate. Blue and red arrows represent the velocity field of CMONOC and AHCORS, respectively. The areas surrounded by dashed line are North China block, Ludong block, and South China block. Error ellipses represent 70% confidence.

If uniform strain exists in the block, the strain rate can be calculated by using Formula (2) given by Savage et al. [33]. In this case, the model is obtained by adding Formula (1) to Formula (2). The velocity caused by the internal strain in the block can be written as

$$
\begin{bmatrix} V_\lambda \\ V_\theta \end{bmatrix} = \begin{bmatrix} R\sin\theta_0\Delta\lambda & R\Delta\theta & 0 \\ 0 & R\sin\theta_0\Delta\lambda & R\Delta\theta \end{bmatrix} \begin{bmatrix} \dot\varepsilon_\lambda \\ \dot\varepsilon_{\lambda\theta} \\ \dot\varepsilon_\theta \end{bmatrix} \tag{2}
$$

where $\lambda$ and $\theta$ correspond to the colongitude and colatitude, respectively, while the $\Delta\lambda$ and $\Delta\theta$ are the colongitude and colatitude differences between the measuring point and the regional geometric center. $R$ is the mean radius of the regional geometric center and $\theta_0$ is the colatitude of the regional geometric center. $V$ and $\dot\varepsilon$ are the velocity component and strain rate component, respectively.

After the GPS horizontal velocity field of a certain region is solved, grid searches or simulated annealing method are then used to invert the fault coupling ratios and slip rate deficit. The coupling ratio is defined as a value between 0 and 1. A value of 1 indicates that the fault patch is fully locked and a value of 0 means that the fault patch is freely creeping. A value between 0 and 1 suggests that the fault is partly locked. The quality of parameter fitting can be evaluated using the reduced $\chi^2$ statistic which is defined as follows [32]:

$$
\chi_n^2 = [\,\sum_{i=1}^{n} \left(\frac{r_i}{f\sigma_i}\right)^2\,]/dof \tag{3}
$$

where $n$ is the number of observed data and $r_i$ is the residual of observed data. For $f$, it represents the error weight factor, which is generally between 1 and 5 [34]. $\sigma_i$ is the standard deviation and dof is the degrees of freedom.

In order to obtain a set of optimal solutions, we need to adjust the size of $f$ repeatedly during inversion to make $\chi_n^2 \approx 1$, so that the model is able to simulate the observed data accurately.

### 3.3. Block Definition and Fault Geometry

According to the geological and geodetic information, Zhang et al. [35] delimited active blocks in China. As a result, our research area is bounded by the central and southern segments of the Tanlu fault zone and divided into three parts: North China block, Ludong block, and South China block. In the inversion process, we assumed that the South China block is an internally stable rigid block, while there is uniform strain in the North China block and Ludong block [2]. The central and southern segments of the Tanlu fault zone have a strike of SSW, a downdip of NW, and the dip angle is fixed at 65° [36]. The fault plane is composed of fifteen nodes along the strike, and the average distance between nodes is about 50 km. Meanwhile, the setting of nodes along the downdip is based on the research results of Li et al. [16]. According to the results of earthquake relocation [7], the depth of earthquakes in the central and southern segments of the Tanlu fault zone are mostly within 30 km, and only a few earthquakes exceed 30 km. Therefore, eight independent nodes are set along the downdip direction, with the depth of 0.1 km, 5 km, 10 km, 15 km, 20 km, 25 km, 30 km, and 35 km. At present, no studies show that there is creeping in the shallow of the central and southern segments of the Tanlu fault zone. Accordingly, a strong constraint with fault coupling ratios of 1.0 is added to the nodes at 0.1 km and 5 km [32], and it is assumed that only free slip exists below 35 km. The coupling ratios of the fault between 0.1–35 km decrease monotonically along the downdip.

## 4. Results

### 4.1. Fault Coupling Ratios

According to the parameter settings above, we utilized the velocity field data of AHCORS and CMONOC to obtain our optimal coupling model. For the preferred model, when the sigma scaling factor f of its horizontal velocity field data is taken as 2.754, $\chi_n^2$ is just equal to 1.000 (the number of observations is 1290, the degree of freedom is 1212). Figure 3 depicts the comparison between observed and predicted GPS velocity field and the velocity residuals for the optimal model. The mean residual of the north and east components is −0.223 and 0.058 mm/y, respectively. This indicates that the fitting result of the model is precise.

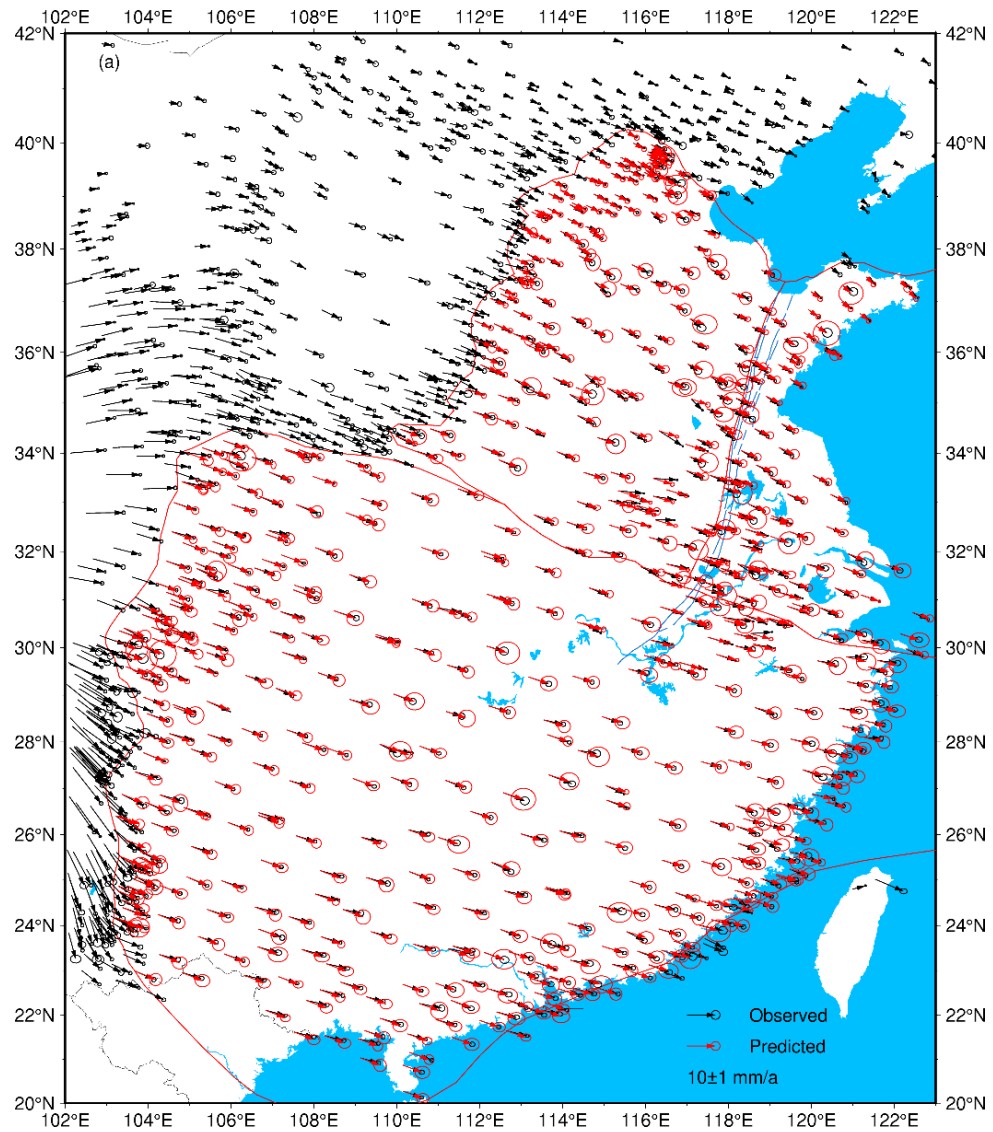

**Figure 3.** *Cont.*

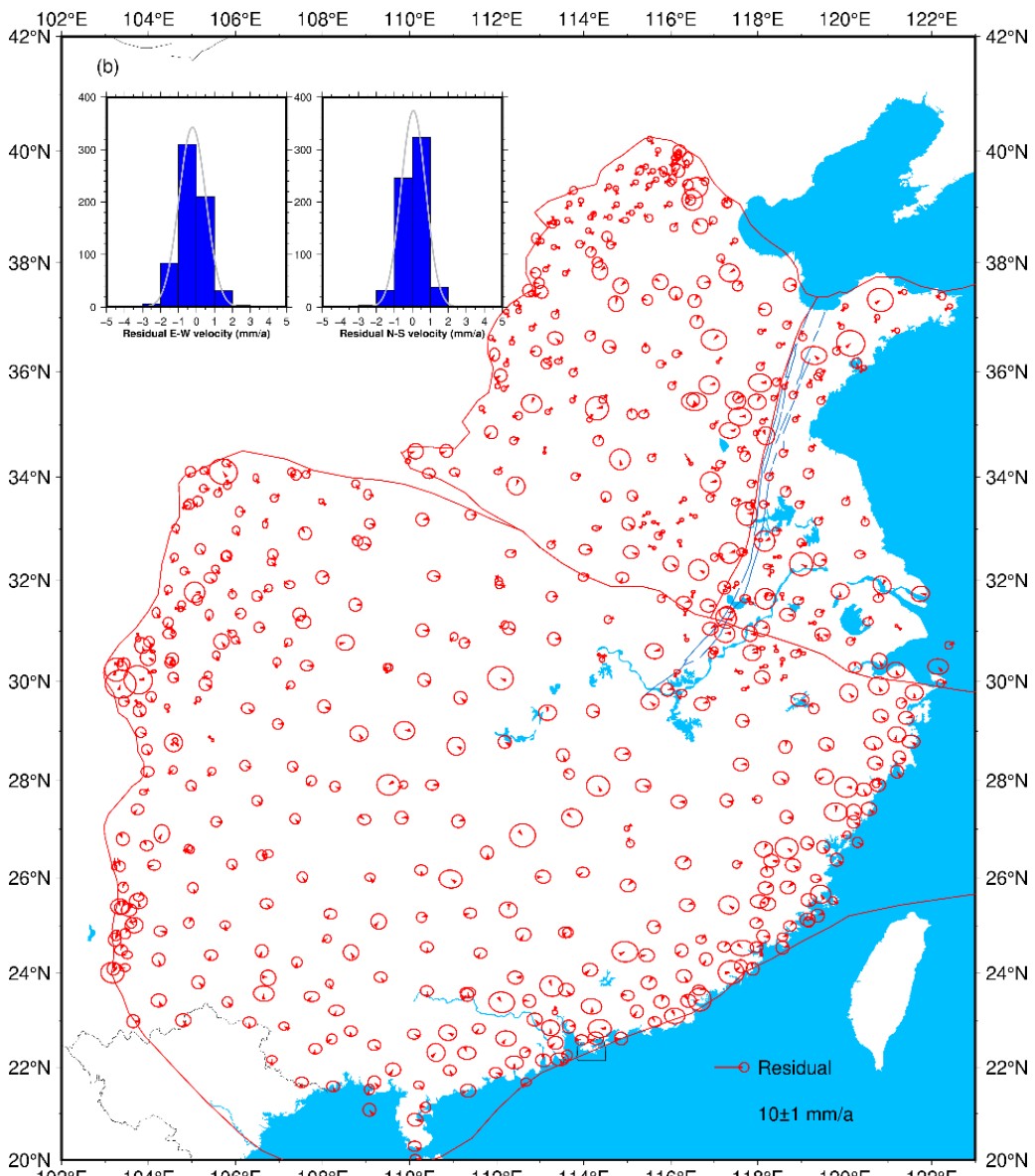

**Figure 3.** Comparison between observed and predicted GPS velocity field (**a**). Black and red arrows are observed velocity and predicted velocity, respectively. GPS velocity residuals distribution for the optimal model (**b**). The images in the upper left corner are the statistical histogram of residuals of the east components and north components, respectively.

The fault coupling distribution on the central and southern segments of the Tanlu fault zone is shown in Figure 4. For the convenience of analysis, the whole fault is divided into three parts from north to south: the Weifang–Tancheng segment, the Tancheng–Jiashan segment, and the Jiashan–Tongcheng segment. They are all located in eastern China, and most places are plains and hills. From Figure 4, lateral variation of coupling distribution can be found along the strike of the Tanlu fault zone. The Weifang–Tancheng segment is in a state of high coupling 26 km below the surface, with coupling ratios above 0.8. Along the downdip, the coupling ratios decrease with the increase of depth. The coupling ratios of 26–30 km change from high coupling to medium coupling, and the coupling ratios are about 0.6. Along the downdip, from 30 km to 35 km, the fault changes from strong coupling to freely creeping. As for the Tancheng–Jiashan segment, its locking depth is 5 km below the surface, which is much shallower than that of the Weifang–Tancheng segment. The vicinity of Tancheng at the north end is in a fully coupling state within the uppermost

26 km below the surface. The epicenter of the 1663 Tancheng $M_s$ 8.5 earthquake is close to this region. The lower edge of the source fault is determined to be about 32 km depth [37], coinciding well with the strong coupling area. For the Jiashan–Tongcheng segment, its middle and north sections are in a medium coupling state. The middle section especially is still in a strong coupling state 25 km below the surface; however, the south section is freely creeping below the depth of 10 km. Generally, compared with the southern segment of the Tanlu fault zone, the northern segment has a higher coupling degree and a deeper locking depth, which means that the strain accumulation in the northern segment is more rapid.

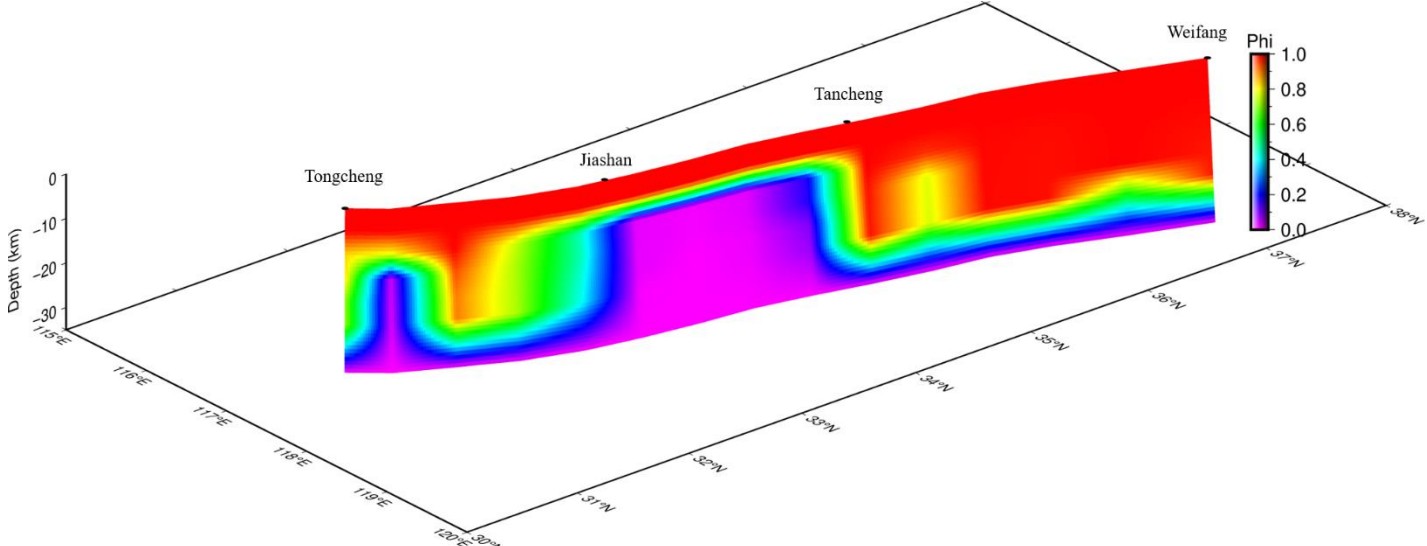

**Figure 4.** The three-dimensional (3D) spatial distribution of coupling ratios of the optimal model inverted by AHCORS and CMONOC velocity field. Purple to red indicates the fault coupling coefficient. The red places indicate that the fault is fully locked, and the purple places mean that the fault is freely creeping; other colors suggest that the fault is partly locked.

### 4.2. Fault Slip Rate Deficit

Figure 5 shows the slip rate deficit of the optimal model on the central and southern segments of the Tanlu fault zone. In Figure 5, the 3D spatial distribution of slip rate deficit is similar to coupling ratios. For the Weifang–Tancheng segment, the slip rate deficit is the largest within 30 km below the surface, ranging from 0.8 to 1.6 mm/a. For the Tancheng–Jiashan segment, the slip rate deficit gradually decreases from north to south, and the rate is between 0.6–1.0 mm/a. As for the Jiashan–Tongcheng segment, the slip rate deficit is between 0.2–0.5 mm/a.

Fault slip rate deficit is calculated by multiplying the slip rate by the coupling coefficient. For the central and southern segments of the Tanlu fault zone, we can conclude, by combining the coupling ratios in Figure 4, that although the segments are fully coupled within 5 km underground, the slip rate deficit at the north is larger than that at the south, indicating that the slip rate at the north is larger. This result is consistent with the conclusion of Guo et al. (2011), that the slip rate of Tanlu fault zone gradually decreases from north to south (1.24–1.06 mm/a) [38].

### 4.3. Velocity Profiles Analysis

It is a conventional method to analyze the relative motion between blocks by GPS velocity profiles across the fault. As shown in Figure 1, we have made three profiles from north to south along the Tanlu fault zone. To facilitate the distinction, we represented them as profile a, profile b, and profile c from north to south, and the velocity profile results are shown in Figure 6. Through the profiles, it is clearer to see some features of the velocity field. For the velocity component parallel to the profile lines, when the slope of the red line

is positive, it indicates extension, while when the slope is negative, it indicates compression. Meanwhile, as for the velocity component perpendicular to the profile lines, positive slopes of blue lines mean left-lateral. On the contrary, negative slopes are right-lateral [28,30,39].

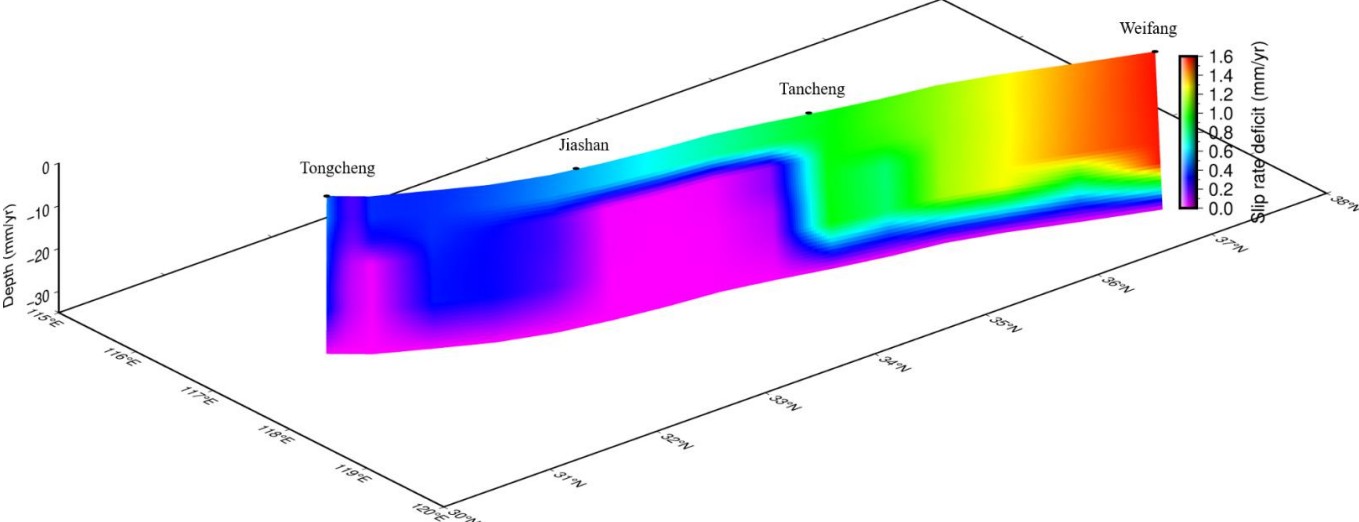

**Figure 5.** The 3D spatial distribution of slip rate deficit of the optimal model inverted by AHCORS and CMONOC velocity field (mm/a). Purple to red indicates the slip rate deficit. The red places indicate the slip rate deficit is large, and the purple places mean that there is no slip rate deficit.

In Figure 6, we compare the observed (points) and calculated (lines) GPS velocity and found that there are a few misfits between observed and calculated velocity results from longer distance to the profiles. However, on the whole, the fitting result is quite accurate. From the profiles, the velocity parallel to the profile lines and perpendicular to the profile lines changes slightly when passing through the fault; it is continuous without any obvious step, indicating that the fault cannot slip freely and is still locked. At the same time, the velocities of profile a, profile b, and profile c, whether parallel or perpendicular to the profile lines, show a gentle negative slope. We also derived the slip rates and root mean square error of the Tanlu fault by calculating the velocity of the profiles, as shown in Table 3. Generally, the velocity parallel to the profiles is between −0.76 and −0.35 mm/a, and the velocity perpendicular to the profiles is between −0.44 and −0.29 mm/a. Hence, it proves that the fault among the profiles is right-lateral and compressive, and the compression component gradually decreases from north to south. From geological studies, Li et al. (2019) suggested that the kinematic characteristics of the Tanlu fault zone are right-lateral and thrust [40], which is consistent with our result.

**Table 3.** Slip rates on the central and southern segments of the Tanlu fault zone (left-lateral with tension is positive).

| Segment | Velocity Parallel to the Profiles/RMSE (mm·a$^{-1}$) | Velocity Perpendicular to the Profiles/RMSE (mm·a$^{-1}$) |
|---|---|---|
| a | −0.76/0.16 | −0.44/0.22 |
| b | −0.57/0.11 | −0.40/0.15 |
| c | −0.35/0.16 | −0.29/0.17 |

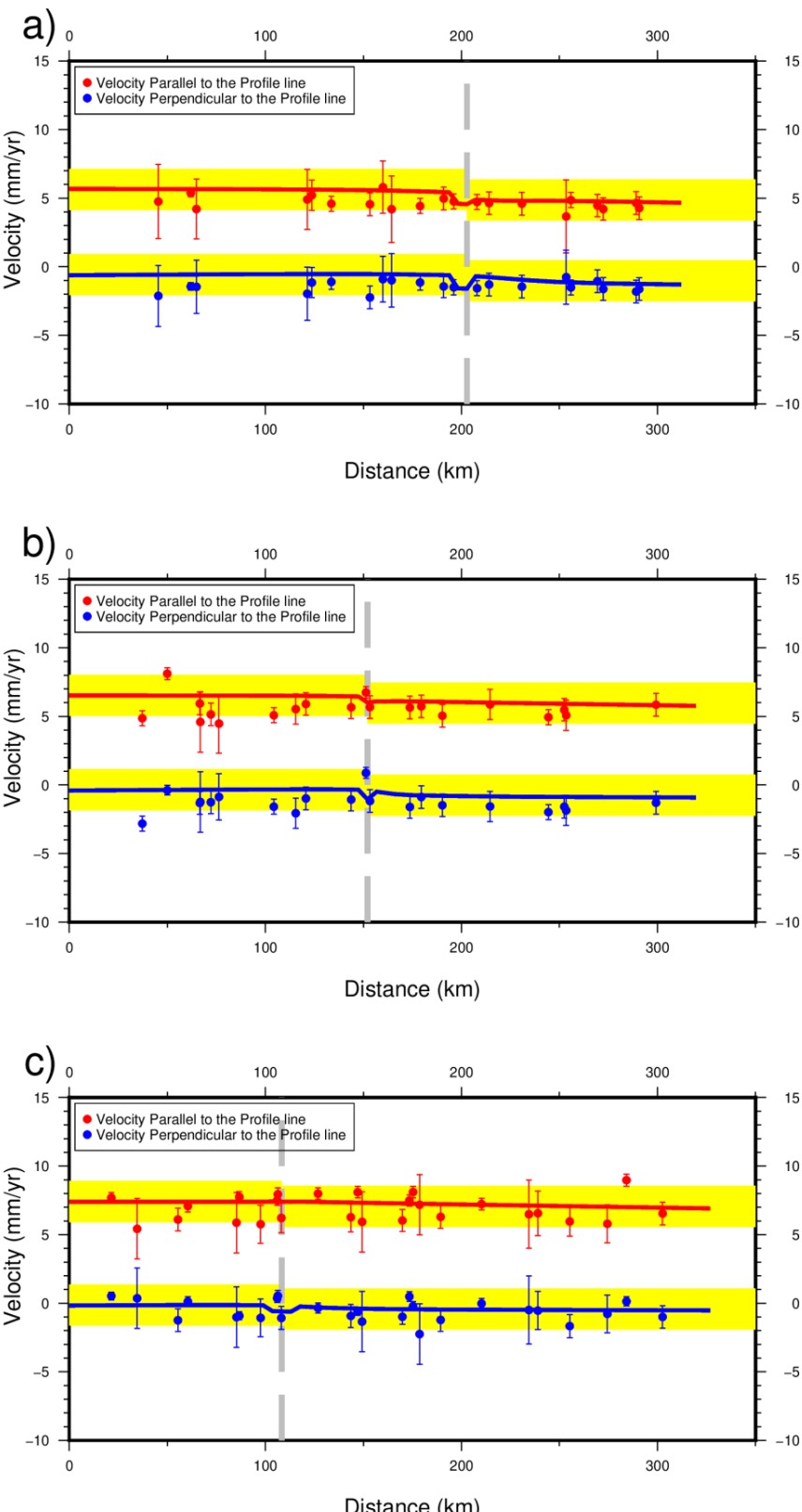

**Figure 6.** GPS velocity profiles from north to south (**a–c**). The length of the profiles is less than 350 km, and the width is 160 km. The red points and lines represent the observed and calculated GPS velocity parallel to the profile lines, respectively. The blue points and lines represent the observed and calculated GPS velocity perpendicular to the profile lines. The gray dashed line represents the Tanlu fault zone, and the median value of the yellow rectangle is the average of the velocities.

## 5. Discussion

### 5.1. Checkboard Tests

We conducted a series of checkboard tests to determine the minimum distance along the strike of the coupling ratios that the GPS velocity field can resolve. First, in the forward modeling, we input the rotation parameters of the blocks to estimate the velocity of each station, and then added the Gaussian noise to obtain a new velocity field. Finally, we used the synthetic velocity field to invert the coupling ratios of the fault and compared it with the forward modeling result [41,42], as shown in Figure 7. We compared the cases where the distance between nodes along the strike is 50–70 km and found that when the distance between nodes is 60 km and 70 km, the input information cannot be recovered well, especially at the depth of 20 km below the surface. In Figure 7f, while the distance between nodes is 50 km, most grid cells are recovered well except for a small part, which may be caused by inhomogeneous distribution of velocity field. As a result, we chose 50 km as the minimum distance between adjacent nodes. From the spatial resolution of the coupling ratios, we can conclude that in order to obtain the optimal inversion results, we need to consider the GPS velocity density.

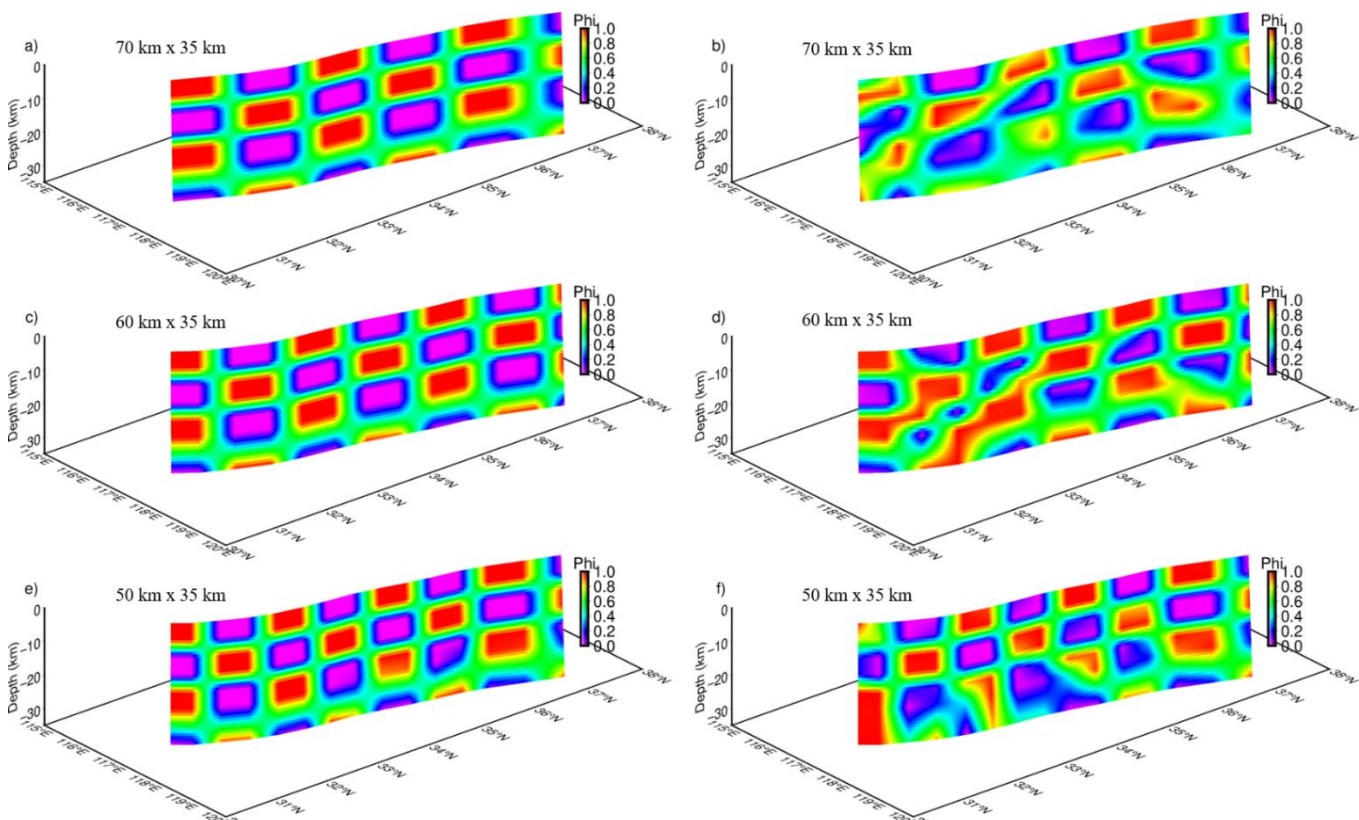

**Figure 7.** Resolution tests for coupling ratios inverted by different distances between adjacent fault nodes. Figures on the left (**a**,**c**,**e**) are 3D distribution of coupling ratios for forward modeling, and on the right (**b**,**d**,**f**) are recovered 3D distribution of coupling ratios using the same inversion strategy.

### 5.2. Comparison with Previous Studies

In order to explore whether near-field data will affect the inversion results of the model, we used only CMONOC velocity field to invert the distribution of fault coupling ratios and slip rate deficit, as shown in Figure 8. Comparing the results of this inversion with the results integrating the near-field data of AHCORS, we found that the fault coupling ratios and slip rate deficit are quite similar. In the Tancheng–Jiashan segment, both of them are in a state of strong coupling 26 km below the surface, and the difference mainly lies in the

Tancheng–Jiashan and the Jiashan–Tongcheng segments, which also exactly corresponds to the position of AHCORS velocity field. In the Tancheng–Jiashan segment, the locking depth of the inversion result for the optimal model is relatively shallower than that of this result. In the Jiashan–Tongcheng segment, the inversion result of the optimal model is strongly locked in the middle section, and the depth can reach 25 km, which is not reflected in this inversion. As for slip rate deficit, except for the Jiashan–Tongcheng segment, other parts are basically the same. Compared with the results of Li et al. (2020) [16], the main difference is also shown in the south of Tancheng. As a result, we believe that this is caused by the integration of near-field data, which significantly affects the inversion results.

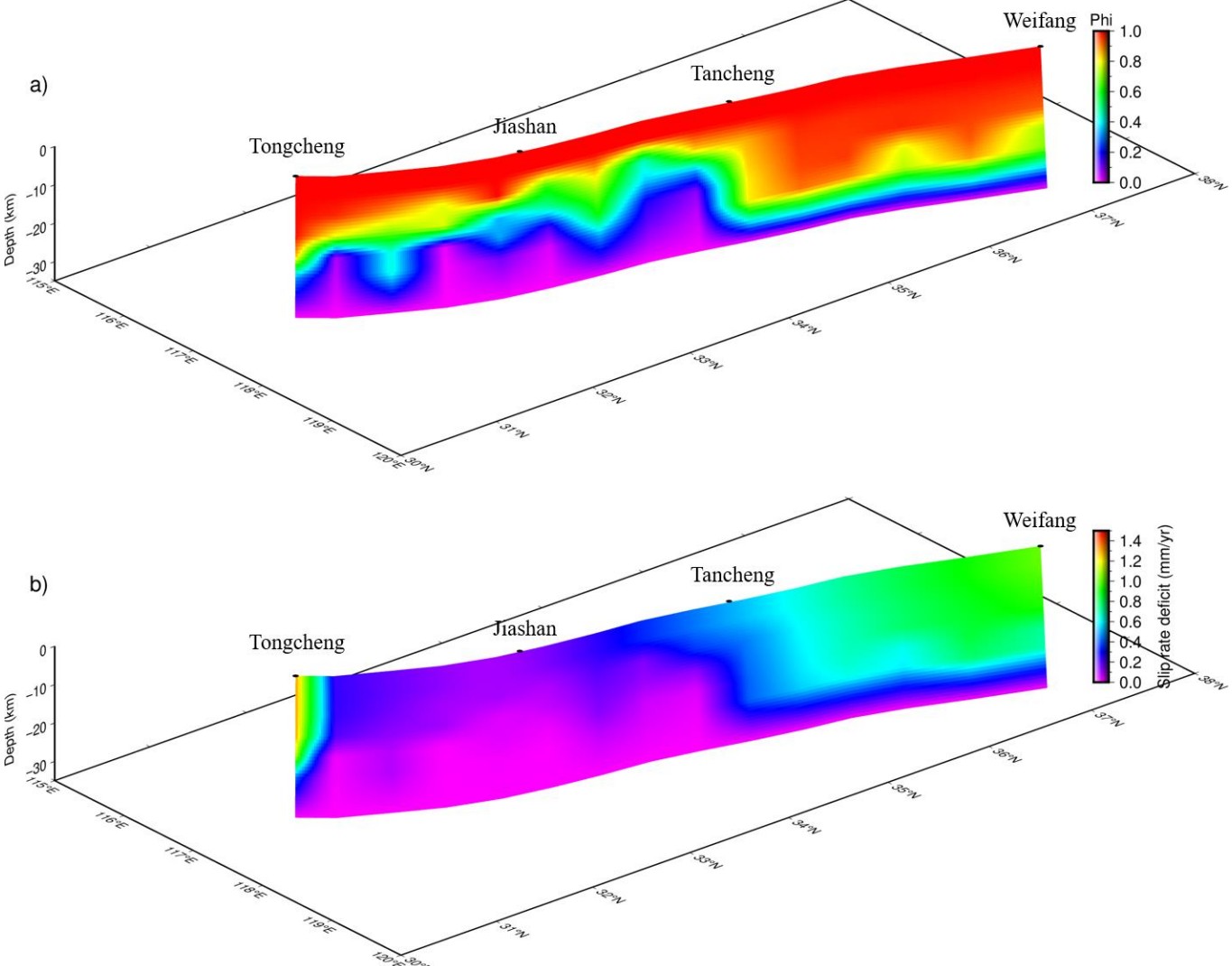

**Figure 8.** The 3D spatial distribution of coupling ratios inverted only by CMONOC velocity field (**a**). The 3D spatial distribution of slip rate deficit inverted only by CMONOC velocity field (**b**). The values in the northern segment are mostly larger than those in the southern segment.

*5.3. Strain Characteristics*

The inconsistency of spatial distribution of the GPS horizontal velocity field is a direct reflection of crustal deformation. Different reference frame will lead to a large difference in velocity field; however, the strain rates are not related to the reference datum, and it is one of the crucial indicators to describe regional surface deformation directly [43–45].

Taking GPS horizontal velocity field as constraints, we used DEFNODE to invert the coupling ratios and slip rate deficit of the Tanlu fault zone. Next, we utilized GPS

horizontal velocity field to calculate regional strain rates, to describe surface deformation characteristics and fault activity. In this paper, we chose the horizontal velocity field from 245 stations including CMONOC and AHCORS in the longitude and latitude range of $113°$ E–$122°$ E and $27°$ N–$38°$ N, and calculated the principal strain rates, dilatation rates, and maximum shear strain rates of the central and southern segments of the Tanlu fault zone and surrounding areas using least-squares collocation method. The horizontal strain rates are shown in Figure 9.

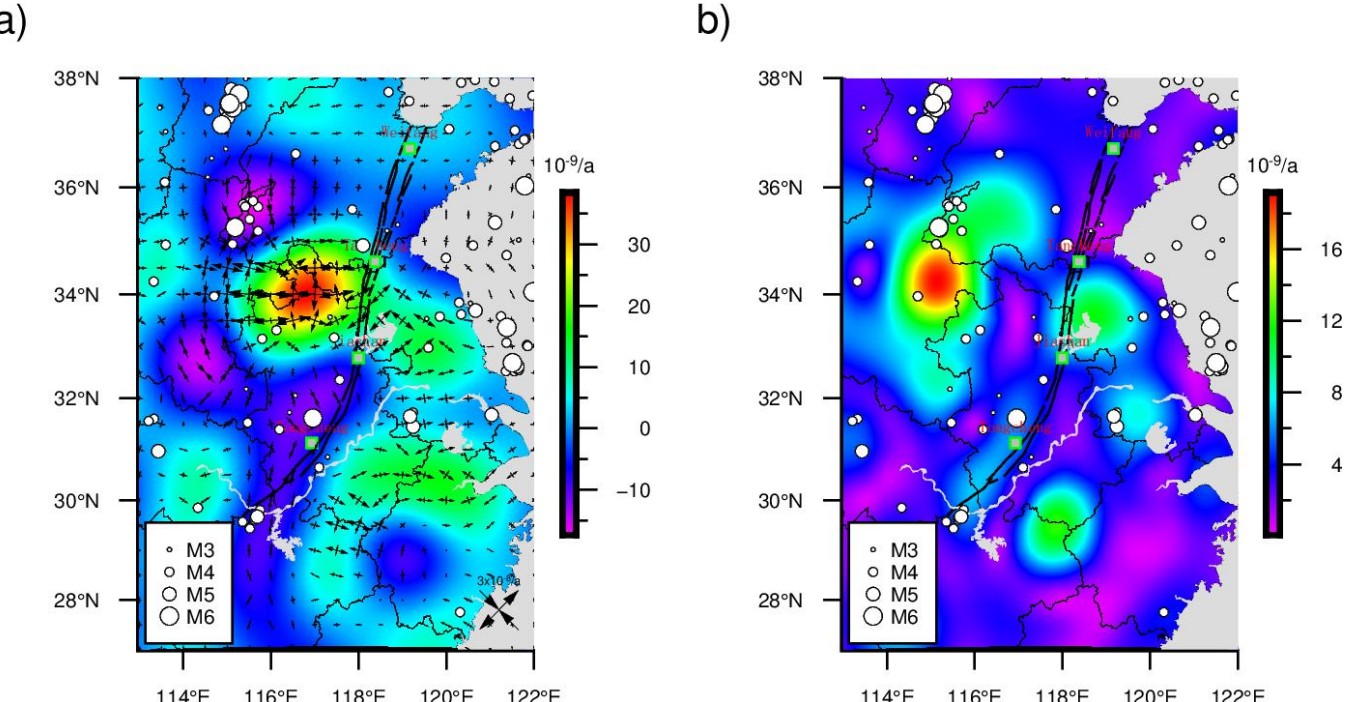

**Figure 9.** Strain rates on the central and southern segments of the Tanlu fault zone. (**a**) Principal strain rates and dilatation rates. Principle strain rates are shown as vector pairs and dilatation rates are shown in background color. Positive dilatation rates show extension, while negative show compression. (**b**) Maximum shear strain rates. White circles show earthquakes with $3.0 \leq M_s \leq 8.5$ from 1 January 1900 to 31 December 2020.

In Figure 9a, the principal strain rates near the Weifang–Tancheng segment of the Tanlu fault zone are almost zero. The Tancheng–Jiashan segment corresponds to the high-value area of the principal strain rates. Bounded by the Tanlu fault zone, the west side shows a nearly E–W and N–S extension, while the east side is a NE–SW or NW–SE extension. The principal strain rates of Jiashan–Tongcheng are relatively small, and the east side is dominated by approximately E–W extension, and the west side is nearly N–S compression. For dilatation rates, the maximum value appears on the west side of the Tancheng–Jiashan segment, which is $3.78 \times 10^{-8}$/a. It is in the north of Anhui Province, where shallow groundwater has been exploited for a long time. Additionally, there are abundant coal resources here, and years of mining have contributed to the subsidence of the area. Therefore, we believe that it is likely to be caused by human activities, which should attract public attention.

In Figure 9b, it can be recognized that the maximum shear strain rates in the central and southern segments of the Tanlu Fault zone are smaller than those on both sides of the Tancheng–Jiashan segment and the east of the southern segment of Tongcheng. Generally, the maximum shear strain rates in the central segment of the Tanlu fault zone are much larger than the southern segment, consistent with the geological results indicating that the activity of the southern segment is weaker than the central segment [22].

*5.4. Implication for Future Seismic Hazard*

Generally, we believe that the occurrence of an earthquake is a process of continuous strain accumulation. When the accumulation reaches the limitation, it will break through the stable state, resulting in sudden rupture of the fault zone [46–48]. Therefore, for the fault zone with high coupling ratios and deep locking depth, this means the accumulation of strain rates, and it is more likely to have a large earthquake in the future. At the same time, slip rate deficit is also an important indicator of the rate of strain accumulation on active faults [49]. Therefore, we have reason to pay attention to the places with high coupling ratios, deep locking depth, and high slip rate deficit or abnormal strain rates on the surface.

For the central and southern segments of the Tanlu fault zone, according to the spatial distribution of coupling ratios, we could find that the Tancheng–Weifang segment is in a state of high coupling 26 km below the surface, with coupling ratios above 0.8. In this segment, there are only two major earthquakes above $M_s$ 8.0, occurring in Anqiu in 70 BC and Tancheng in 1668. Notably, the focal depth of the 1663 Tancheng $M_s$ 8.5 earthquake is 32 km, which is deeper than the strong coupling depth of 26 km based on our inversion result. It indicates that the locking depth was at least 32 km before the earthquake, and the locking degree has not returned to its previous state, which may be the reason why there has been no earthquake with a magnitude lager than $M_s$ 8.5 in this region so far. From geological studies, Li et al. (2019) considered that the age of the latest paleoseismic event was about $12.8 + 4.0/-3.7$ ka through paleoseismic trough and AMS $- ^{14}C$ dating method [40]. As a consequence, the recurrence interval of long-period large earthquakes is the main feature of Late Quaternary activity in the central and southern segments of Tanlu fault zone. In general, the fault coupling ratios and slip rate deficit of the northern segment are larger than those of the southern segment, which indicates that the northern segment is more prone to generate large earthquakes. Although it has been more than 350 years since the 1668 Tancheng earthquake, the Weifang–Tancheng segment still deserves our attention.

For the distribution of the strain rates, the area with large principal strain rates also corresponds to the high-value area of the dilatation rates, and the interface area between extension and compression is always accompanied by a sudden change in the direction of the principal strain rates. The places where the strain rates are abnormal may have potential for disaster. They provide an important reference for us to prevent seismic hazard.

## 6. Conclusions

Using the data of AHCORS and CMONOC stations, we inverted the coupling ratios, slip rate deficit, and velocity profiles by DEFNODE on the central and southern segments of the Tanlu fault zone. We found that slip rate deficit and locking depth in the north is higher and deeper than that in the south and it is more likely to produce strain accumulation. In particular, the locking degree of the Tancheng has not been restored to the state before the 1668 Tancheng earthquake, so its adjacent region has not experienced a large earthquake for a long time. Based on geological studies and historical large earthquakes, it can be seen that the Tanlu fault zone is characterized by long-period recurrence interval of large earthquakes. By comparing the coupling ratios and slip rate deficit, the fault shows that it has high coupling ratios within 5 km under the surface; however, the slip rate deficit in the north is larger. Hence, the slip rate in the north is also larger than that in the south. Subsequently, we analyzed three velocity profiles across the fault zone. The result shows that the velocity parallel to the profiles is between $-0.76$ and $-0.35$ mm/a, and the velocity perpendicular to the profiles is between $-0.44$ and $-0.29$ mm/a, which indicates that the fault is right-lateral strike-slip and compressive. Finally, we used least-squares collocation to calculate the strain rates. The results suggest that where the principal strain rates are large, the value of dilatation rates will also be large. The interface of extension and compression is always accompanied by sudden change of direction of principal strain rates. These places with abnormal strain rates have the potential for disaster.

While we used AHCORS data to obtain a high-precision velocity field on the central and southern segments of the Tanlu fault zone, the measurements of the AHCORS, as is

known, could be affected by various types of uncertainties and inaccuracies that arise from different causes. It would be necessary to evaluate the fuzziness [50] and jumps of data [51], which play an important role in estimating trends and forecasting in the future work. At the same time, we believe that the setting of fault geometry can be further improved, such as setting different dip angles at different depths of the fault. Additionally, only GPS data were used to invert the coupling pattern of the Tanlu fault zone in this study. In the future, a refined coupling model jointly constrained by multisource data (e.g., InSAR and GPS data) could help us to better understand the strain accumulation and seismic risk in the Tanlu fault zone.

**Author Contributions:** Conceptualization, H.C., T.T. and S.L.; methodology, H.C. and S.L.; software, H.C.; validation, H.C. and T.T.; formal analysis, H.C.; investigation, H.C.; resources, H.C.; data curation, H.C.; writing—original draft preparation, H.C.; writing—review and editing, H.C., T.T. and S.L.; visualization, H.C.; supervision, X.Q. and Y.Z.; project administration, T.T.; funding acquisition, T.T. All authors have read and agreed to the published version of the manuscript.

**Funding:** This work is funded by Open Research Fund Program of Hunan Province Key Laboratory of Safe Mining Techniques of Coal Mines (Hunan University of Science and Technology) under Grant E22015, and also supported by the Natural Science Foundation of Anhui Province, China under Grant 1808085MD105. This work is supported by the National Natural Science Foundation of China under Grant 42004001.

**Acknowledgments:** We are grateful to the Anhui Bureau of Surveying and Mapping for providing the raw GNSS observation data. We sincerely thank Min Wang for publishing the velocity field data of CMONOC.

**Conflicts of Interest:** The authors declare no conflict of interest.

**Abbreviations**

The following abbreviations are used in this manuscript:

| | |
|---|---|
| GPS | Global Positioning System |
| InSAR | Interferometric synthetic aperture radar |
| AHCORS | Anhui Continuously Operating Reference System |
| CMONOC | Crustal Movement Observation Network of China |
| IGS | International GNSS Service |
| VMF1 | Vienna Mapping Function 1 |
| FES2004 | Finite 79 Element Solutions 2004 |
| SOPAC | Scripps Orbits and Permanent Array Center |
| ITRF2008 | International Terrestrial Reference Frame 2008 |
| 3D | Three-dimensional |

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
