# Peer review of "Interseismic Fault Coupling and Slip Rate Deficit on the Central and Southern Segments of the Tanlu Fault Zone Based on Anhui CORS Measurements"

_remotesensing, doi:10.3390/rs14051093_

Round 1
Reviewer 1 Report
*) Please explain more clearly in the Introduction of the paper the motivation that prompted the Authors to use the method of placing least squares.
*) In Section 2.2 little emphasis has been placed on formalizing units of measurement. Please fill this gap.
*) Some captions of the images are not self-explanatory. Please fill in this small gap.
*) The measurements of the global positioning system (GPS), as is known, could be affected by various types of uncertainties and inaccuracies that arise from different causes. So, with this premise, it would be necessary to evaluate the fuzziness contained in the available data to fully understand if the data processing must be operated by a fuzzy processor. Obviously, such an activity goes beyond the work carried out by the Authors who, however, I ask to insert a sentence in the text that highlights this possibility by putting the following relevant work in the bibliography:
doi: 10.1515/phys-2020-0159
This paper that I recommend to insert presents an innovative approach to evaluating the fuzziness in data through the formulation of specific indices. Even if the approach is applied to a civil engineering problem, the transversality of the procedure can be fully applied to the problem studied by the authors.
Reviewer 2 Report
Reviewer’s Report on the manuscript entitled:
Interseismic Fault Coupling and Slip Rate Deficit on the Central and Southern Segments of the Tanlu Fault Zone Based on Anhui CORS Measurements
The authors processed the GPS measurements of Anhui Continuously Operating Reference System (AHCORS) between January 2013 and June 2018 and derived a high-precision velocity field in the central and southern segments of the Tanlu fault zone. The topic and results are interesting, but the presentation must be improved. Below, please see my comments.
The Introduction needs further improvement. There is no mention of using InSAR data for monitoring earthquakes. Please discuss this and also add the following recent articles on interseismic fault coupling and slip rate deficit:
https://doi.org/10.3390/rs13163333
https://doi.org/10.3390/rs10060943
Lines 10, 11, etc. Please use a consistent style when you define the abbreviations. In here you used: global positioning system (GPS) with lower case and then you defined Anhui Continuously Operating Reference System (AHCORS) with upper case. So, please say Global Positioning System (GPS) and check everywhere else in the manuscript. Please ensure that all the abbreviations are defined the first time they appear in the manuscript. Also, please add an abbreviation table at the end of the manuscript, listing all the acronyms used in the manuscript.
Line 36. What is the unit of magnitude here? Is it Richter? Please write it here and also in the caption of Figure 1 and elsewhere.
Line 86. It should be “least-squares” not “least square”. Please correct throughout the manuscript.
Line 88. Please add a paragraph describing how the rest of the paper is organized. Something like: The rest of the paper is organized as follows. In Section 2, the study region, datasets, and methodology are described. Section 3 demonstrates the results…etc.
Figure 1. Please insert a scalebar in kilometers and a color bar for the scene background. Is it elevation? Also, what are (a), (b), (c) inside the figure?
Figures 4, 5 ,7, 8. The way that you illustrated the spatial distribution is misleading. As it looks like a plane in a 3D space. I suggest you demonstrate either a 3D surface or a 2D like Figure 1.
Figure 6. Please insert the x-axis and y-axis labels and their units. Also, please insert the legend.
Figure 6. The following article describes a software, namely, Jumps Upon Spectrum and Trend (JUST) that is designed to estimate trends and jumps in time series by considering the error bars and can be used for forecasting as well. Please make a paragraph in Introduction and/or Discussion to discuss this or as for the future work:
https://doi.org/10.1007/s10291-021-01118-x
Line 307. Please use the capital letter for “Figure”, not “figure”. Please check and correct elsewhere.
In the Conclusion, please also discuss the limitation of this study and future directions.
Thank you for your contribution
Regards,
Reviewer 3 Report
Article Review – Remote sensing
Title reviewed ms: Interseismic Fault Coupling and Slip Rate Deficit on the Cen-2 tral and Southern Segments of the Tanlu Fault Zone Based on 3 Anhui CORS Measurements
Author(s) Tingye Tao, Hao Chen, Shuiping Li 1, Xiaochuan Qu, and Yongchao Zhu
MS No.: remotesensing-1582945
General comment
Dear Authors,
This paper used the global positioning system (GPS) measurements of Anhui Continuously Operating Reference System (AHCORS) between 2013 and 2018 to derive a high-precision velocity field in the central and southern segments of the Tanlu fault zone, a major strike slip curve fault in Eastern China. The manuscript presents an interesting topic, which should catch the attention of the readers of Remote Sensing. As far as I know, the manuscript has not been published previously. The title is conforming with the contents of the Ms and the approach and results and conclusions intelligible from the abstract alone. The structure of the paper lacks of some descriptions related to the geology of the study area and although is quite sharp, it needs a general rearrangement. The introduction need of some rewriting to assist a broader audience. A geological setting should be included. Results need to be more carefully explained and cleaned of interpretations, which would suite well the discussion section.
Some specific comments
Before going into regional tectonics, please state in the beginning what is interseismic fault coupling and slip rate defict and what is the geological significance that this two topics may have in best understanding the seimic behavior of large strike-slip faults crosscutting the whole crust such as the Tanlu fault zone. In the introduction you should possibly also mention similar studies on similar structures elsewhere on Earth and how the issue was approached. Then you can say what best suits your case and why.
A paragraph on the Tectonic setting of Eastern China is here needed as it is relevant for a good geologic understanding of the later presented results and for the inferences of the discussion.
Among the topics this paragraph should treat, there should be information on the:
- setting of the crust;
- geological history with particular reference to the Tanlu fault zone and the previous studies dealing with it.
- geological evidences of lateral variation along the fault trace and the Author's geodynamic and tectonic interpretation of such differences.
Some parts can be taken already from the introduction which could be more focused on the general issue a strike-slip fault of this size bears when dealing with interseismic fault coupling and slip rate defict.
Suggestions for improving technical points have been provided with detailed comments that will help preparing the final version of the manuscript. When reviewing, please consider my comments and the proposed changes in the attached pdf file. After random checks if the literature, I found that the reference list corresponds to the cited papers in the text. No excess of self-citations was found. An overall English check is recommended for a final version.
Notwithstanding these moderate corrections, it deserves publication.
Best wishes

Round 2
Reviewer 1 Report
Reference 62 is written incorrectly.Please correct it with the following correct version:
Versaci, Mario, Angiulli, Giovanni, di Barba, Paolo and Morabito,
Francesco Carlo. "Joint use of eddy current imaging and fuzzy similarities
to assess the integrity of steel plates" Open Physics, vol. 18, no. 1,
2020, pp. 230-240. https://doi.org/10.1515/phys-2020-0159
Reviewer 2 Report
I would like to thank the authors for addressing most of my comments. Please see below my remaining comments.
- If you want to keep Figures 4,5,7,8 with their current style, then please at least insert the z-axis label and unit (is it in meters?) and also the color bar unit and label for them.
- Please note that references [51] to [63] are not referred to in the manuscript and they must be referred to. For example, in line 461, you can add the last reference [63] right after "...jumps of data [63]".
- Please check the references one by one to ensure that their style and format are according to the MDPI guidelines and to ensure their correctness and to ensure they are all properly referred to in the body of the manuscript.
- Finally please carefully proofread the article before publication if accepted by the editor.
Thank you for your contribution
Regards,
Reviewer 3 Report
Dear Authors,
I find your answers satisfying. In my opinion this paper is now suitable to be published.
Best wishes,
Luca Cardello
Author Response
Dear Reviewer,
We sincerely appreciate very much for your positive and constructive comments and suggestions on our manuscript.
Best wishes,
Hao Chen